# Relationship between Meteorological and Air Quality Parameters and COVID-19 in Casablanca Region, Morocco

**DOI:** 10.3390/ijerph19094989

**Published:** 2022-04-20

**Authors:** Mohamed Khalis, Aly Badara Toure, Imad El Badisy, Kenza Khomsi, Houda Najmi, Oumnia Bouaddi, Abdelghafour Marfak, Wael K. Al-Delaimy, Mohamed Berraho, Chakib Nejjari

**Affiliations:** 1International School of Public Health, Mohammed VI University of Health Sciences (UM6SS), Casablanca 82403, Morocco; atoure@um6ss.ma (A.B.T.); ielbadisy@um6ss.ma (I.E.B.); k.khomsi@gmail.com (K.K.); obouaddi@um6ss.ma (O.B.); cnejjari@um6ss.ma (C.N.); 2General Directorate of Meteorology, Casablanca 20000, Morocco; najmi_houda@yahoo.fr; 3National School of Public Health, Rabat 10000, Morocco; ab.marfak@gmail.com; 4Laboratory of Health Sciences and Technology, Higher Institute of Health Sciences, Hassan 1st University of Settat, Settat 26000, Morocco; 5Herbert Wertheim School of Public Health and Human Longevity Science, University of California, San Diego, CA 92093, USA; waldelaimy@health.ucsd.edu; 6Department of Epidemiology, Faculty of Medicine and Pharmacy of Fez, Sidi Mohamed Ben Abdellah University, Fez 30000, Morocco; maberraho@gmail.com

**Keywords:** COVID-19, air pollutants, air quality, meteorological parameters, time-series analysis, generalized additive model, Morocco

## Abstract

The aim of this study was to investigate the relationship between meteorological parameters, air quality and daily COVID-19 transmission in Morocco. We collected daily data of confirmed COVID-19 cases in the Casablanca region, as well as meteorological parameters (average temperature, wind, relative humidity, precipitation, duration of insolation) and air quality parameters (CO, NO_2_, 03, SO_2_, PM10) during the period of 2 March 2020, to 31 December 2020. The General Additive Model (GAM) was used to assess the impact of these parameters on daily cases of COVID-19. A total of 172,746 confirmed cases were reported in the study period. Positive associations were observed between COVID-19 and wind above 20 m/s and humidity above 80%. However, temperatures above 25° were negatively associated with daily cases of COVID-19. PM10 and O_3_ had a positive effect on the increase in the number of daily confirmed COVID-19 cases, while precipitation had a borderline effect below 25 mm and a negative effect above this value. The findings in this study suggest that significant associations exist between meteorological factors, air quality pollution (PM10) and the transmission of COVID-19. Our findings may help public health authorities better control the spread of COVID-19.

## 1. Introduction

Coronavirus disease (COVID-19) was discovered in December 2019 in China in the city of Wuhan and was declared a pandemic by the World Health Organization (WHO) on 11 March 2020, given its high transmission and rapid progression. By 21 March 2020, more than 170 countries had reported an outbreak of COVID-19. As of 4 February 2022, WHO reported that there were 386,548,962 confirmed cases of COVID-19, including 5,705,754 deaths worldwide [1]. In Morocco, the first case of SARS-CoV-2 was detected on 2 March 2020. As of 31 December 2020, the number of active COVID-19 cases was 24,301 and the number of deaths were 7388. About 1861 new cases were reported on 31 December 2020, 40% of which were in the Region of Casablanca [2].

Independent of vaccination status, virus transmission is affected by a number of factors, including but not limited to: host behavior; host defense mechanisms; virus infectivity; population density; and environmental factors. Among these, the environment is primarily responsible for the dispersal, dissemination, and infectivity of the novel coronavirus SARS-CoV-2 and its variants. In fact, environmental factors have long been recognized as potential contributors to viral transmission, as is the case for the influenza virus, SARS and the dengue [3]. It is well-established that the main route of transmission of SARS-CoV-2 is through coughing droplets from carrier individuals. The mobility of these droplets was shown to be affected by weather parameters such as wind, temperature and humidity [4]. Therefore, the effect of these parameters on viral transmission has been the subject of several research investigations. Similarly, previous systematic reviews suggest that air quality indicators such as PM2.5 and NO_2_ and to a lesser extent PM10, contribute to the transmission and lethality of COVID-19 [5].

The association of SARS-CoV-2 transmission with meteorological parameters such as humidity, temperature, winds and sunlight has been examined in previous studies [4,6,7,8,9,10,11,12]. The results from recent systematic reviews and meta-analyses show significant associations between COVID-19 cases and weather factors [13,14]. However, findings reported by orther studies indicated a weak association between COVID-19 viral spread and weather parameters [15]. Previous findings also show a bidirectional relationship between SARS-CoV-2 and the environment [16]. Several other studies also indicated that air pollution may be a vector of infection and an important risk factor of severity for COVID-19 [17,18,19,20]. Many studies were also conducted in African counries such as Mozambique [21], Nigeria [22], and Algeria [15]. A study conducted in 52 African states, including Morocco, using a Generalized Additive Model (GAM) found that temperature and humidity were negatively associated with new cases of COVID-19 whereas average wind speed had an inverse effect on COVID-19 growth in Africa [23].

To the best of our knowledge, this is one of the first studies to explore the association between meteorological parameters, air quality parameters and the number of daily cases of COVID-19 in Morocco. Environmental factors and the other factors of transmission of COVID-19 are expected to be different across regions given the difference in population dynamics, infrastructure, as well as public health and preventive measures. Identifying key environmental factors affecting the transmission of COVID-19 in a country like Morocco can be relevant to other middle- and low-income countries suffering from COVID-19 and future potential pandemic waves, and could help inform public health policies and improve disease surveillance and control. The aim of this study was to investigate the dose-effect relationship relationship between meteorological parameters, air quality and daily confirmed COVID-19 cases in the Casablanca region, Morocco, using General Additive Models (GAM).

## 2. Materials and Methods

### 2.1. Data Collection

Data on daily confirmed COVID-19 at Casablanca region for the period of 2 March 2020, to 31 December 2020, was collected from the Ministry of Health of Morocco. Daily meteorological parameters and daily air quality parameters for the same period were obtained from the General Directorate of Meteorology (GDM) of Morocco. The daily meteorological parameters included in this study were: temperature (°C); humidity (%); precipitation (mm); wind (m/s); and total duration of the insolation (exposure to the sun) (h). The air quality parameters used in this study included: sulfur dioxide (S02, µg/m^3^); nitrogen fioxide (NO_2_); suspended particulate matter with an aerodynamic diameter of less than 10μm (PM10); varbon monoxide (CO); and ozone (O_3_). The air quality parameters, for the study period, were measured in five monitoring stations set up in the Casablanca region (Figure 1). The five stations were selected among all the existing stations because of the data availability. Every station is a background one. It monitors five air pollution indicators: SO_2_, NO_2_, O_3_, CO, and PM10. Sampling and measurement are performed instantly using automatic devices that automatically introduce the gas sample and record the output. These devices are referred to as continuous analyzers. The recorded outputs are then averaged every 15 min and archived in the air quality database. At the GDM, the control of air quality data is performed according to the rules and recommendations from the Agency for the Environment and Energy Management [24]. It is carried out according to two main stages: (1) the automatic prevalidation that is a systemic process performed, according to special rules, at the level of the measuring device, the acquisition system, and the central concentration station; and (2) the expert validation by a qualified person according to two mandatory steps. The first step is the technical validation, which consists of checking the conformity of the whole measuring system. The second step is the environmental validation which consists of a cross-sectional investigation of the relevance and consistency of the obtained data. As for meteorological data control, it is carried out at the GDM according to the recommendations of the WMO to National Meteorological and Hydrological Services [25]. Overall, to ensure data quality, the GDM adopts a clear process in the framework of its ISO 9001 certified management system. The process proactively prevents potential risks that could cause data quality deterioration, with a well-designed climate data management system and well-trained personnel aware of data quality culture. After collecting the data, data experts at the GDM identify and formulate data problems and implement corrective actions, through regular reviews and continuous improvement processes. Missing data for the whole data series were imputed by the mean of the observed values per day for each indicator.

### 2.2. Statistical Analysis

Data collected from the five stations were combined and the overall average of each parameter was calculated. We used General Additive Models (GAM) to explore the dose-effect relationship between meteorological factors, air quality parameters, and daily confirmed COVID-19 cases in the Casablanca region. The GAM is an extension of the Generalized Linear Model with a linear predictor including a sum of smooth functions of covariates. The general equation of the model is as follows: gi (μY) = f_1 (X1_) + f_2 (X2_) + … + f_k (Xk_), where gi (μY) = E (Y) corresponds to the expected value of response variable Y, and fn () are the smooth functions of the covariates Xk. The concept of additivity comes from the sum of smooth functions which can be illustrated as follows: gi (µY) = Σ(fi (Xi)).

The GAMs model allows for a great flexibility in the estimation of the parameters because each covariate can have its own distribution. This helps to reduce the error terms and increase the precision of the model parameters estimation [26]. In this study, temperature, precipitation, humidity, wind, insolation, PM10 and O_3_ were adjusted as covariates. The general equation of our final model is as follows:E(Yt) = α + s(Temperature) + s(Precipitation) + s(Humidity) + s(Wind) + s(Insolation) + s(PM10) + s(O_3_) + lag1(y)
where the response variable E (Yi) is the number of new daily confirmed COVID-19 cases on day t; α is the intercept; and s (.) is the thin plate regression spline function of covariates temperature, insolation, PM10, O_3_, precipitation, humidity and wind.

Correlation coefficients were calculated for all the data matrix using the Spearman rank correlation. The Spearman’s non-parametric test was used to estimate whether this correlation is statistically significant or not. Since distributions of meteorological factors are not normally distributed, Spearman coefficients were preferred over the Pearson correlation coefficients.

To account for the serial correlation in our time series data, a lag1 (Y − t − 1) was introduced indicating the confirmed COVID-19 cases on day t − 1. In our descriptive analysis, we identified strong correlations between some pollutants. To avoid multicollinearity, the covariates of our model were selected based on the literature and by applying a cutoff of rho = 0.4 on our correlation matrix. This cutoff corresponds to a moderate correlation level [27].

Four models were estimated, and the final model was selected using two metrics: AIC (the Akaike Information Criterion) and the Adjusted R2. A diagnosis of the final model has been made by verifying the smoothness selection optimization, checking the residual plots and testing whether the basis dimension choices are adequate. Finally, Relative Risks (RRs) were calculated to assess the relationship between meteorological parameters, air quality, and daily COVID-19 cases. All statistical analyses were performed using the R statistical software [28]. The estimation of the model was done with using the mgcv package [29]. Relative risks were computed with dlnmTS package [30].

## 3. Results

### 3.1. Descriptive Statistics of the Parameters

Table 1 presents descriptive statistics of meteorological parameters, air quality parameters, and daily confirmed COVID-19 cases for the period of 2 March 2020, to 31 December 2020.

During the study period, the total number of confirmed COVID-19 cases was 172,746 with a daily average of 566 cases [Min = 0; Max = 2517]. The daily averages of meteorological parameters were 20.13 (±3.5) °C, 0.83 (±3.6) mm, 71.64 (±7.1)%, 11.83 (±2.9) m/s and 8.74 (±3.0) h for temperature, precipitation, humidity, wind and insolation, respectively. The daily averages of air quality parameters were 0.14 (±0.09) mg/m^3^ for CO, 18.41 (±12.7) µg/m^3^ for NO_2_, 33.18 (±14.8) µg/m^3^ for O_3_, 46.4 (±24.5) µg/m^3^ for PM10, and 3.86 (±3.0) µg/m^3^ for SO_2_. Figure 2 shows the time-series distribution of daily confirmed COVID-19 new cases in the Casablanca region from 2 March 2020 to 31 December 2020. The number of confirmed cases reached its maximum (2517) on 11 May 2020. Figure 3 presents the time-series distribution of meteorological and air quality parameters in the Casablanca region at the period of the study.

### 3.2. Spearman Correlations of Meteorological and Air Quality Parameters, and Daily Confirmed COVID-19 Cases

Table 2 presents Spearman correlations of meteorological and Air Quality Parameters, and Daily Confirmed COVID-19 Cases. Moderate negative correlations were observed between wind and CO (ρ = −0.54, *p* < 0.001), wind and NO_2_ (ρ = −0.54, *p* < 0.001). While positive correlations were observed between CO and NO_2_ (ρ = 0.9, *p* < 0.001), CO and SO_2_ (ρ = 0.72, *p* < 0.001), NO_2_ and PM10, and NO_2_ and SO_2_. The rest of the variables had weak correlations.

Regarding the correlation with Daily COVID-19 Cases, wind (ρ = −0.36, *p* < 0.001) and insolation (ρ = −0.3, *p* < 0.001) had a significant negative correlation with new daily COVID-19 cases. While CO (ρ = 0.58, *p* < 0.001), NO_2_ (ρ = 0.6, *p* < 0.001), O_3_ (ρ = 0.42, *p* < 0.001), PM10 (ρ = 0.4, *p* < 0.001) and SO2 (ρ = 0.46, *p* < 0.001) had a significant positive correlation with new confirmed COVID-19 cases. On the other hand, temperature, precipitation and humidity had non-significant correlation with new confirmed COVID-19 cases, unlike wind (ρ = −0.36, *p* < 0.001) and insolation (ρ = −0.3, *p* < 0.001) that had significant negative correlations.

### 3.3. GAM Model

Table 3 shows the results of the GAMs model. The effect of smooth terms temperature, precipitation, humidity, wind, insolation, PM10, and O_3_ on the evolution of the daily confirmed COVID-19 cases was significant. All those factors had a non-linear relationship with daily confirmed COVID-19 (EDF > 1). The model’s explanatory power was substantial (R2 = 0.73; Deviance explained = 0.8).

Figure 4 illustrates the shape of the exposure-response curves that represent the relationship of temperature (°C), precipitation (mm), humidity (%), wind (m/s), insolation (h), PM10 (µg/m^3^) and O_3_ (µg/m^3^) with the daily confirmed COVID-19, independent of the other variables in the models as covariates.

Overall, PM10 and O_3_ showed an increasing curve with the number of confirmed COVID-19 per day. Positive trends were observed for wind and humidity above certain values, 20 m/s for wind and 80% for humidity. Additionally, temperatures above 25 °C had a negative trend with the number of COVID-19 cases. Also, insolation showed a clear increasing curve above the value of 9 h. While the curve of Precipitation was volatile below 22 mm and decreasing above the same value.

Figure 5 shows the smoothed relationship in the relative risk scale. The RRs for one unit change in meteorological parameters and air quality parameters was associated with a change in the number of daily confirmed COVID-19. For temperature, we note a fluctuating RR below 25 °C, and a significant decrease above that value. The precipitation level had an inconstant effect on the daily trend of COVID-19 cases below 25 mm but decreased significantly above this value. For humidity, a significant increase in RR was observed from 80% onwards. The RR of wind remained close to 1, dropped slightly starting at 17.5 m/s wind speed and then increased significantly from wind speeds of 26 m/s onwards. Similarly, the RR for Insolation dropped significantly starting from 10 h or more exposure to sun. O_3_ had a RR under 1 below 40 µg/m^3^ but started fluctuating above this value. Finally, the RR of PM10 remained close to 1, however, a singular increase was observed starting from 100 µg/m^3^.

## 4. Discussion

This study examined the relationship between meteorological parameters, air quality parameters and the number of daily confirmed COVID-19 cases in Casablanca, Morocco from 2 March 2020 to 31 December 2020. Our results indicate the presence of non-linear relationships between daily confirmed COVID-19 and meteorological factors namely temperature, precipitation, humidity, wind and insolation as well as air quality parameters, mainly PM10 and O_3_. An increase in the average of temperature, precipitation and insolation may decrease the number of daily COVID-19 cases, while an increase in humidity, wind, PM10 and O_3_ may increase the number of daily COVID-19 cases. In our GAM model, the deviance explained by meteorological and air pollution parameters on daily COVID-19 cases was 0.8, close to the result found in a similar study [31].

Our results showed a significant negative association of temperature above 25 °C with daily confirmed COVID-19. Similar results were observed in several other studies performed in 9 of the most infected cities in the world [32]. Investigations conducted in China also reported a significant negative association between temperature and transmission of COVID-19 above 24 °C [33], 20 °C [34] and 10 °C [35], respectively. Our results are consistent with a study conducted in 52 African countries which reported a significant negative association of mean temperature with COVID-19 cases [23]. Other studies conducted in Pakistan [8] and Taiwan [36] reported a significant positive association between temperature and viral transmission. A study in Bangladesh found that a 1 °C increase in temperature was related to a 14.2% drop in confirmed cases [37], this number was 4.09% in Brazil [38]. This may be explained by the weakening of host immunity brought about by cold weathers which leads to an increased susceptibility to infection, as is the case for seasonal viral infections such as the Influenza Virus. Considering the similarities between the transmissibility of SARS-CoV-2 and the Influenza Virus, we believe the mechanisms to be fairly comparable [39]. However, other studies conducted in Canada, China and Indonesia reported no significant association between temperature and COVID-19 transmission [40,41,42]. In contrast, a study in neighbouring Algeria found a weak correlation between temperature and COVID-19 cases [15]. Temperature and solar radiation have been the most studied meteoroloical parameter and several interpretations were proposed to explain the inverse effect on COVID-19 cases. Some authors suggest that solar irradiation–which leads to the rise in temperature–exposes the virus to UV radiation which may damage its lipid membrane and limit its infection potential [39]. Other authors suggest that UV radiation may improve immunity by increasing vitamin D levels thereby preventing transmission and the onset of symptoms associated with SARS-CoV-2 [43]. Further studies are needed to conclude about the effect of this parameter on the COVID-19 transmission and its underlying mechanisms.

With regards to precipitation, our study showed a negative significant association with daily confirmed COVID-19 above 25 mm. Studies in China also reported a positive significant association below 30 mm [33]. Positive correlation, between precipitation and COVID-19 transmission was identified in a global study including all countries affected by COVID-19 on the 1 December 2019 [44]. However, other studies in Iran [45] and Indonesia [41] found no association between precipitation and COVID-19 transmission.

Our study findings indicate a positive association between humidity above 80% and daily COVID-19 cases. Similar findings were observed in Singapore (80 ± 4%) and China (50–80%) [33,46]. Opposing results were found in one study in Mainland China where humidity above 70% was inversely associated with transmission of SARS-CoV-2 [34]. In fact, each 1% increase in humidity was shown to decrease the number of daily cases by 11 to 22% was concluded in a study on COVID-19 transmission in Mainland China [35]. These findings may be explained by an increase of suspended matter in the atmosphere when the weather is dry, creating favorable conditions for virus resistance and its transmission. These mechanisms are comparable to those of the influenza virus [44]. Similarly, in dry indoor settings where humidity is below 40%, transmission of SARS-CoV-2 was higher compared to places where the humidity was 90% [38]. However, some contrasting results were reported in Africa where no significant association between humidity and the exposure-response curve of COVID-19 was observed [23]. Further studies are needed to investigate the mechanism by which humidty affects the viral spread of SARS-CoV-2.

Our findings show a positive association between the wind above 26 m/s and daily COVID-19 cases. Similar results were obtained in other studies in China [34], United States of America [47] and Turkey [48]. These findings may be attributed to the fact that wind is responsible for propagation of airborne particles [49] and the modulation of the dynamics of various pathogens and vectors [48]. However, some contradicting findings were observed in a study in Iran which reported a negative association between wind and transmission in provinces with a high infection rate and low wind speeds [45]. The findings in the present study can be explained by the fact that in outdoor settings, high wind speed leads to removal of the airborne virus particles, while in indoors, a high velocity improves ventilation. This is consistent with the recommended ventilation measures for workspaces aimed at SARS-CoV-2 prevention and control [50].

In this study, the duration of insolation beyond 10 h was inversely associated with the daily COVID-19 cases. Similar findings were reported in an Iranian study showed an inverse association between the duration of insolation beyond 4 h and cases of COVID-19 [45]. Scientific literature suggests that viral exposure to ultraviolet irradiation for 60 min in a culture medium leads to the destruction of viral infectivity to levels that cannot be detected. Consequently, longer exposure to solar radiation in outdoor settings means more exposure to ultraviolet light that may lead to a reduction in the transmissibility of COVID-19 [51].

It is worth noting that the association observed between viral transmission of SARS-CoV-2 and weather parameters may also be attributed to changes in human behaviour that accompany seasonal changes. These behaviours can inhibit or faciitate viral transmission. For example, the school and university academic year which often takes place during cooler weather, which means that children and students spend more time indoors during the day and go back to their family households later in the evening, thereby facilitating the spread of the virus. Additionally, the time spend by people in indoor settings during extreme hot or cold weathers increases. Nevertheless, better understanding the effect of weather parameters on the spread of SARS-CoV-2 requires taking into consideration other non-envrimental factors [52].

The current study showed that PM10 levels above 100µg/m^3^ were positively associated with daily cases of COVID-19. These findings are in agreement with those from other studies in China [53,54], Italy [7,55], USA [56] and France [57]. Contrasting results were reported in a China study, in which an increase in PM10 concentrations was correlated with a decrease in the incidence of COVID-19, as opposed to PM2.5 [58]. The association showed in this study and from other studies may be due to PM acting as carriers that viral particles latch on. Additionally, air pollutants have been shown to increase the permeability of mucosa and oxidative stress, decrease defense mechanisms such as phagocytosis, and enhance COVID-19 transmissibility by weakening the immune system [9].

Our findings indicated that O_3_ levels below 40 µg/m^3^ may be positively associated with the daily COVID-19 cases but this relationship is fluctuating and therefore not consistent. A study conducted by Zhu et al. in China reported a positive association between O_3_ and the newly COVID-19 cases, and that 10 µg/m^3^ increase was associated with 4.76% (95% CI: 1.99 to 7.72) [59]. Also, a study conducted in Thailand showed that O_3_ was significantly related to increased risk of respiratory hospital admission [60]. This may be due to an increased permeability of epithelial cells in the lungs and changes in airway microbiota caused by the inhalation of O_3_, both of which lead to increased vulnerability to respiratory infections including SARS-CoV-2 [61,62]. However, several studies indicated that O_3_ emissions can inhibit COVID-19 and even lead to the inactivation of the virus at high O_3_ doses [63,64]. Therefore, further investigations that examine the effect of ozone on viral transmission are needed.

To the best of our knowledge, this is one of the first studies to explore the relationship between environmental factors and daily COVID-19 cases in Morocco. However, it comes with certain limitations; first, this study did not take into consideration information on demographic and socio-economic characteristics, population mobility, immunity level and geographic density. This limitation restricted quantifying with precision the effect of meteorological parameters and air quality on COVID-19 transmission. Second, the study was performed by analyzing data in the aftermath of the national state of emergency, results could potentially have been influenced by the change in people’s behaviors and the precautionary measures imposed by the government. Third, even if the statistical model that has been chosen (GAMs) is the most popular model for complex relationship, causality cannot be clearly observed. We do not imply there is a cause–effect relationship, and the ecological correlations by their nature are more hypothesis-generating.

## 5. Conclusions

Our study showed significant associations between meteorological parameters and air quality parameters and the daily COVID-19 cases. The results are consistent with numerous other studies. The findings in this this study will allow public health authorities to plan and recast public health measures aimed at curbing the spread of the virus accordingly. Furthermore, taking into account the complex interactions between the level of transmission of viruses and the impact of environmental factors is essential to building a sustainable, reactive and relevant epidemiological surveillance system. Based on these results, further in-depth studies should take into account other regions, levels of pollutants, and demographic characteristics to better understand the interaction between environmental factors and mechanisms of COVID-19 transmission.

## Figures and Tables

**Figure 1 ijerph-19-04989-f001:**
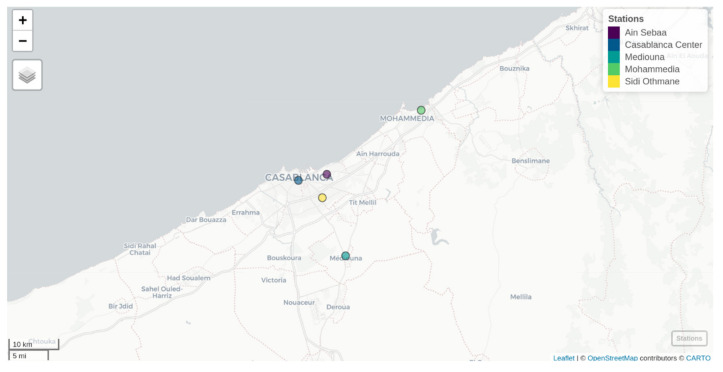
Location map of the meteorological stations included in this study.

**Figure 2 ijerph-19-04989-f002:**
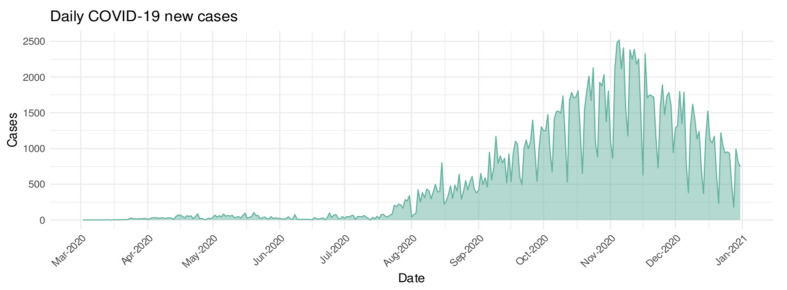
Time series distribution of daily COVID-19 confirmed cases in Casablanca region (from 2 March 2020 to 31 December 2020).

**Figure 3 ijerph-19-04989-f003:**
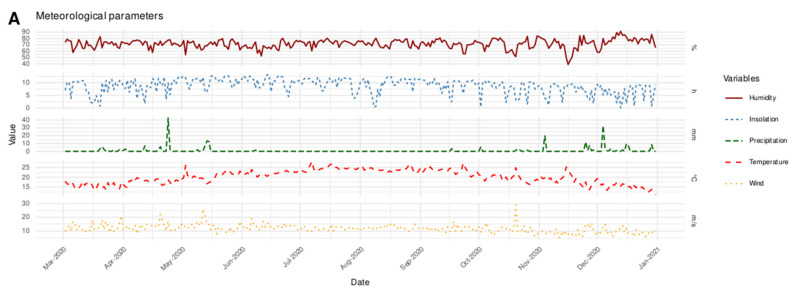
Time series distribution of meteorological (**A**) and air quality (**B**) parameters in Casablanca region (from 2 March 2020 to 31 December 2020).

**Figure 4 ijerph-19-04989-f004:**
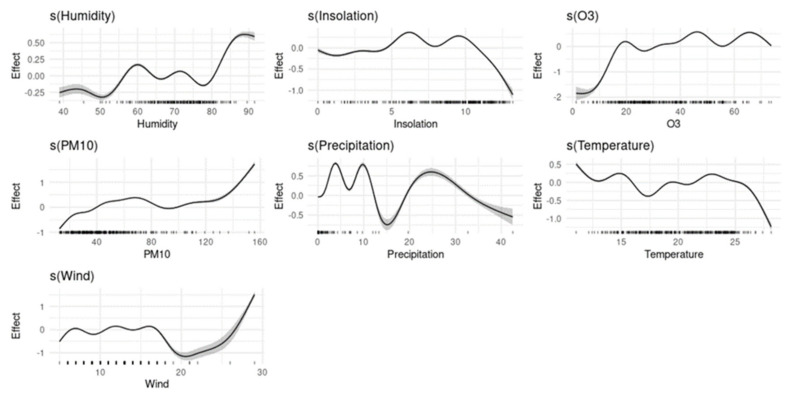
Exposure−response relationship between temperature, humidity, precipitation, wind, insolation, O_3_, PM10, and daily confirmed COVID−19 cases in Casablanca region from 2 March 2020 to 31 December 2020.

**Figure 5 ijerph-19-04989-f005:**
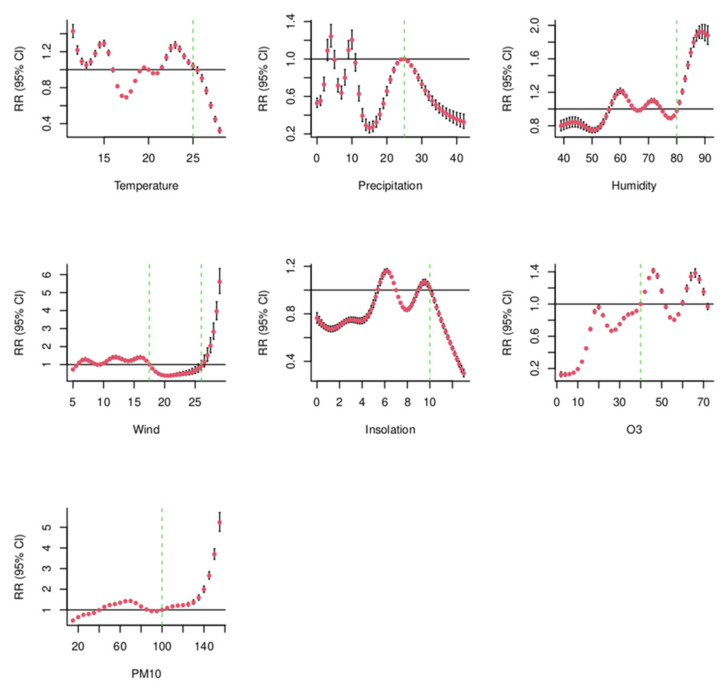
Risk Ratios with 95% confidence intervals increase of daily confirmed COVID-19 associated to change in temperature, humidity, precipitation, wind, insolation, O_3,_ and PM10.

**Table 1 ijerph-19-04989-t001:** Descriptive statistics for meteorological parameters, air quality parameters and daily confirmed COVID-19 in Casablanca region for the period of 2 March 2020 to 31 December 2020.

Variables	Minimum	25%	Median	75%	Maximum	Mean	SD
Temperature (°C)	11.05	17.35	20.35	23.35	28.15	20.13	3.58
Precipitation (mm)	0.00	0.00	0.00	0.00	42.50	0.83	3.68
Humidity (%)	39.00	68.00	72.50	76.50	91.50	71.64	7.17
Wind (m/s)	5.00	10.00	12.00	13.00	29.00	11.83	2.97
Insolation (h)	0.00	7.20	9.40	10.90	13.20	8.74	3.01
CO (mg/m^3^)	0.02	0.08	0.13	0.19	0.46	0.14	0.09
NO_2_ (µg/m^3^)	1.80	8.46	15.29	25.50	62.88	18.41	12.72
O_3_ (µg/m^3^)	1.32	22.63	29.42	44.96	73.72	33.18	14.83
PM10 (µg/m^3^)	12.59	30.10	41.42	55.65	155.75	46.40	24.50
SO_2_ (µg/m^3^)	0.79	1.84	3.00	4.47	21.23	3.86	3.08
COVID-19 cases	0.00	28.00	221.00	1024.00	2517.00	566.38	680.17

**Table 2 ijerph-19-04989-t002:** Spearman correlations of meteorological and Air Quality Parameters, and Daily Confirmed COVID-19 Cases, Casablanca region (from 2 March 2020 to 31 December 2020).

Variables	Cases	Temperature	Precipitation	Humidity	Wind	Insolation	CO	NO_2_	O_3_	PM10
Temperature (°C)	−0.18 **									
Precipitation (mm)	0.09	−0.17 **								
Humidity (%)	−0.02	−0.16 **	0.12 *							
Wind (m/s)	−0.36 ****	0.13 *	0.14 *	−0.16 **						
Insolation (h)	−0.32 ****	0.29 ****	−0.20 ***	−0.19 ***	0.21 ***					
CO (mg/m^3^)	0.56 ****	−0.09	−0.09	−0.06	−0.54 ****	−0.09				
NO_2_ (µg/m^3^)	0.65 ****	−0.29 ****	−0.04	−0.08	−0.54 ****	−0.22 ***	0.90 ****			
O_3_ (µg/m^3^)	0.28 ****	0.33 ****	−0.03	0.14 *	−0.10	0.06	0.13 *	0.07		
PM10 (µg/m^3^)	0.41 ****	0.10	−0.09	−0.09	−0.39 ****	−0.07	0.76 ****	0.73 ****	0.16 **	
SO_2_ (µg/m^3^)	0.42 ****	−0.04	−0.08	−0.33 ****	−0.34 ****	−0.06	0.72 ****	0.78 ****	0.04	0.73 **

**** *p* < 0.0001; *** *p* < 0.001; ** *p* < 0.01 ; * *p* < 0.05.

**Table 3 ijerph-19-04989-t003:** Generalized additive model results.

Variables	EDF	*p*-Value
Temperature (°C)	8.99	*p* < 0.001
Precipitation (mm)	8.75	*p* < 0.001
Humidity (%)	7.99	*p* < 0.001
Wind (m/s)	8.91	*p* < 0.001
Insolation (h)	8.95	*p* < 0.001
PM10 (µg/m^3^)	8.98	*p* < 0.001
O_3_ (µg/m^3^)	8.97	*p* < 0.001

AIC = 51,787.52; Deviance explained = 0.80; R2 = 0.73.

## Data Availability

The datasets used in this study are available from the corresponding author on reasonable request.

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
