# Peer review of "Relationship between Meteorological and Air Quality Parameters and COVID-19 in Casablanca Region, Morocco"

_ijerph, 2022, doi:10.3390/ijerph19094989_

Round 1
Reviewer 1 Report
A very interesting study on relationships between Covid and atmospheric parameters.
I have to clearly state that my opinion is (as many) " correlation is not causation". But the manuscript makes a good job in reporting 'data' as they are and proposes explanations not beyond the findings. Thus, in my opinion, it is a very honest representation of what was observed.
The manuscript is very well presented and of interest to a wide audience.
A complete introduction, a very good presentation of the methodological approach, and a clear presentation of results with a discussion strictly related to findings, That it is very good.
As the authors clearly stated causality cannot be clearly observed, this is a very honest statement, and in the conclusions they draw a path to utilise the study. I really appreciate this.
I have only one concern that rise because of a review study I have conducted: here the covid cases are just a number, but we know that the spread is also a function of human behaviour linked to the kind of work, to the age and others. I really appreciate if the authors can include a statistic related to population hitten by covid as divided by age and activities such as schools, industry, drivers, market employers in order to understand that the spread was ubiquitary and not linked to specific habits of the population in order to feel myself more confident that the results are not only a mere correlation. Just implement the work with a table and a short description.
After that to me the paper is worthy of publication.
Reviewer 2 Report
Meteorological indicators, air quality and daily COVID-19: a correlation study at Casablanca Region, Morocco
General comments
The authors have explored the relationship between atmospheric parameters and COVID-19 in Morocco. This is an interesting study that has the potential to spur other research.
Title:
(1) indicators: temperature, wind speed are not indicators but parameters.
(2) correlation study: authors did more than a correlation study in this manuscript. Why limit the title to just a fraction of the analysis?
I recommend that the title be change to: Relationship between meteorological and air quality parameters and COVID-19 in Casablanca region, Morocco
Abstract:
Line 22: In this retrospective study,… Remove “retrospective”
Introduction
Line 42-43: The figure presented for the number of cases is not true. Do the authors intend to refer to February 4, 2022?
Line 55 – 59: Authors need to do more than summarize all research into one sentence. What methods were used in these studies? What are the basic differences, apart from locations, between what has been done and what you are doing? Which of these studies used correlation and GAM? This paragraph should be expanded to highlight key results especially in Africa.
Line 62: Correlation studies between atmospheric parameters and COVID-19 have been conducted over Africa. Authors should do more research on countries like South Africa, Nigeria, etc.
Materials and Methods
Line 79 – 80: How were these meteorological and air quality parameters measured? Which equipment were used? What was the sampling rates? Was the daily averaging done with sums or means? Any conversion done? Can the data be assessed anywhere?
Line 88 – 89: The authors need to do more than this. How many missing points were there in each data? Clearly state these values to place the data in context. This statement is not clear. Was the mean of the whole data used to replace the missing values? Why not replace with 5 days running mean?
5 stations were described but results for a single unspecified location. Where the data for all the locations combined? If yes, this should be stated.
Results
The figures can do with improved quality.
Table 1: Was the N column obtained before or after filling the missing values? Since the values are all the same, I don’t believe the column is necessary.
Table 2: COVID-19 cases can be added to the correlation matrix, thereby eliminating the need for Table 3.
Discussion
It is not enough to just present the results, possible rationale behind the results should also be presented. For example, how does insolation increase the number of COVID-19 cases? Does it affect the behaviour of people or the virus itself? It has been posited that PM10 can act as carriers of the virus, what about O3 and other gases? Are they suitable for the virus to ride on? Is it possible that the correlation is the other way round, that is – the COVID-19 cases and associated non-pharmaceutical intervention affects the air quality parameters? Precipitation data are problematic to work with due to the zero values. How does this affect the results obtained?
Line 307: This is not the first study of this nature in Africa. Authors should prove beyond reasonable doubt that no similar studies have been found to justify this statement.
Authors should consider these studies
Cambaza, E. M., & Viegas, G. C. (2020). Potential impact of temperature and atmospheric pressure on the number of cases of COVID-19 in Mozambique, Southern Africa. Journal of Public Health and Epidemiology, 12(3), 246-260.
Adekunle, I. A., Tella, S. A., Oyesiku, K. O., & Oseni, I. O. (2020). Spatio-temporal analysis of meteorological factors in abating the spread of COVID-19 in Africa. Heliyon, 6(8), e04749.
Okoh, D., Onuorah, L., Rabiu, B., Obafaye, A., Audu, D., Yusuf, N., & Owolabi, O. (2022). An application of artificial intelligence for investigating the effect of COVID-19 lockdown on three-dimensional temperature variation in equatorial Africa. Geoscience Frontiers, 13(2), 101318.
Ogunjo, S., Olaniyan, O., Olusegun, C. F., Kayode, F., Okoh, D., & Jenkins, G. (2022). The role of meteorological variables and aerosols in the transmission of COVID‐19 during harmattan season. GeoHealth, e2021GH000521.
Reviewer 3 Report
The paper "Meteorological Indicators, Air Quality and Daily COVID-19: a Correlation Study at Casablanca Region, Morocco" deals with the effects of meteorological parameters and air pollutants on the number of COVID-19 cases in the period from March to December 2020. The topic is interesting and relevant to the purpose of the journal, however, it lacks the coverage of many aspects and is scientifically weak, considering that the study of correlations is not sufficient to describe the causal relationships properly. Indeed why the key environmental factors are supposed to be influential on the number of COVID-19 cases should be clarified. Therefore, I think much work needs to be conducted prior to publication in a journal such as IJERPH.
My major concern relates to:
1. The introduction is too short, there is a lack of information on the environmental aspect of the pollutants considered. Also, the purpose of the work should be better defined: How the statistical investigation was conducted and the cause-effect relationship underlying the study.
2. The use of "wind speed" should be preferred to "wind" alone.
3. Section 2.1 as presented is not necessary to readers, site selection can be included in a single section with data collection, expanding the description of the 5 stations with information on the instruments used, temporal resolution (are data acquired on an hourly basis and averaged on a daily basis?).
4. Section 3.1: Weather parameters and pollutants should be provided as mean and sd in the text, and a thorough revision of units should be conducted (e.g. precipitation in m/m?; insolation in h?).
5. Section 3.2: Why the authors chose Spearman's non-parametric test should be clarified. It was not even mentioned in section 2.3
6. Is Figure 2 reporting the average of the values at the 5 stations? It is not very clear. The use of a secondary axis might also make the image easier to read.
7. In Table 2 the authors could report in bold the significant coefficients and avoid paragraph 3.2 or expand it with the outcomes obtained from the significance of the coefficients. The same goes for paragraph 3.3, I think it is not enough to list the significant correlations.
8. Figure 3 is described unclearly in the text and caption, it is not enough to write exposure-response curves, and the writing "s(Parameter, EDF)" can be more explicit for readers. Also, the scale on the y-axis is incomplete.
9. A similar argument applies to Figure 4
10. The sentence at lines 242-244 should be explained or a reference provided
Round 2
Reviewer 1 Report
The Authors have reported properly possible limitations of their findings due to human behaviour. In my opinion, the provided changes performed to the paper are a clear indication to readers and, in this form the paper is worthy of publication
Author Response
We thank the reviewer for this positive comment.
Reviewer 2 Report
The authors have sufficiently improved on the quality of the manuscript
Author Response

(The authors gave the same response as above.)

Reviewer 3 Report
The revised version of the manuscript is much improved from that originally submitted. The text has been expanded with the missing information, but also the sections already present in v1 have been improved.
However, one point in my opinion should be improved, which is still about Figure 2: In the review of v1, I clearly did not mean a three-dimensional graph, but the addition of a secondary y-axis, which for example would help the reading of precipitation, insolation or wind trends (panel b). Also, in panel c, the parameter CO appears as a straight line, without variability, as well as SO2. Providing a secondary axis would give a way to read these parameters, with a different scale. As it is, the figure is not representative of the actual variability.
Furthermore, the addition of the units of measure next to each parameter in the legend and the use of more abbreviated writing for dates (e.g. "Apr-20") is recommended to improve the graphics of the figure.
Author Response
We thank the reviewer for this positive feedback.
The comment has been addressed in the Figure 3 of this revised manuscript.